# Effectiveness evaluation of connected and automated vehicles' driving loop: Node weights and driving reliability

**Junlian Yan[1,2], Daowen Zhang**[1,2,3]**, Qirui Luo[4], Jixiang Yang[1,2], Lei Xu[1,2], Hao Xu[1,2], Yihong Zhang[5]***

1 School of Automobile and Transportation, Xihua University, Chengdu, China, 2 Sichuan New Energy Vehicle Intelligent Control and Simulation Test Technology Engineering Research Center, Chengdu, China, 3 Sichuan Traffic Judicial Appraisal Center, Chengdu, China, 4 Dongfang Electric Bulk Cargo Logistics Co., Ltd. Chengdu, China, 5 Sichuan 3trees Paint Co., Ltd, Chengdu, China

* 0119910025@mail.xhu.edu.cn

## Abstract

As an emerging development trend in the automotive industry, the construction of the network model and the effectiveness evaluation of the driving loop for Connected and Automated Vehicles (CAVs) are of significant importance. The objective of this paper is to construct a network model of the driving loop for CAVs and evaluate the effectiveness of the model, thereby optimizing system performance and enhancing driving safety and reliability. In this study, by integrating the driving process of CAVs and introducing the concept of the Observation, Orientation, Decision, and Action (OODA) loop, a network model of the driving loop for CAVs was established, enabling effective modeling of the complex driving process. For effectiveness evaluation, a method is proposed. This method measures the importance of nodes using the Interpretive Structural Model (ISM) and complex network theory, considers driving reliability through the fuzzy evaluation method, and comprehensively determines the node weights of the network model. Subsequently, by utilizing the node weights to enhance the information entropy model, a scientific evaluation of the CAVs' driving loop effectiveness is achieved. Through comparisons and validations across several scenarios, it has been demonstrated that this method can be effectively applied to the planning, modeling, evaluation, and optimization of CAVs network models.

## 1 Introduction

With the continuous advancement of technology, including the rapid development of sensor technology, communication technology, and artificial intelligence, the Connected and Automated Vehicles (CAVs) have emerged as a crucial direction for the transformation and upgrading of the automotive industry. Additionally, the effectiveness evaluation of CAVs during the driving process has emerged as one of

**Data availability statement:** All relevant data are within the manuscript and its Supporting Information files.

**Funding:** This study was supported by: Project of National Automobile Accident In-depth Investigation System (Grant Number: 202248) awarded to Daowen Zhang (Xihua University). Research on the key technologies of in-depth investigation and accident reconstruction of intelligent vehicle (Grant Numbers: 282023Y-10408 / 2023MK185) awarded to Daowen Zhang (Xihua University). Research on Scene Reconstruction of Traffic Accidents Involving Intelligent Vehicles and Its Application in Safety Assessment (Grant Number: KF202211) awarded to Daowen Zhang (Xihua University). Research on the Identification of Freight Traffic Safety Hazard Points and the Design Method of Safety Facilities Based on Big Data (Grant Number: 2025CSLJT3-573) provided by DONGFANG ELECTRIC BULK CARGO LOGISTICS CO., LTD. to Qirui Luo. The specific roles of the funded authors are detailed in the 'Author Contributions' section.

**Competing interests:** The authors have declared that no competing interests exist.

the prominent topics in automotive research. Vehicle effectiveness evaluation is a process that comprehensively and systematically analyzes and assesses a vehicle's capability, efficiency in accomplishing various tasks, and the extent to which goals are achieved in specific usage scenarios. With the continuous evolution of systems science and complex network theory, their applications in effectiveness evaluation are becoming more and more extensive.

In previous research on vehicle network modeling, Yang L et al. [1] proposed a vehicle network model based on the peer-to-peer (P2P) network, aiming to enhance the network's fault tolerance and ensure the stability of the network system. Ma J et al. [2] developed a stochastic electric vehicle network model that incorporated environmental costs and proposed an algorithm for solving this network model. Tu Q et al. [3]proposed a reliability-based equilibrium model for electric vehicle networks, which was employed to predict the traffic flow patterns in the road networks of both electric vehicles and gasoline vehicles. Xu Q et al. [4] constructed and validated a hybrid traffic network model in which human-driven vehicles and CAVs coexisted.

Regarding the research on vehicle effectiveness, Zhao J et al. [5] established an overall subjective evaluation model associated with three objective indicators, leveraging the Probabilistic Neural Network (PNN) to assess the dynamic performance of vehicles. Researchers such as Wang B [6] endeavored to utilize the characteristic indicators of wheels to assess the stability performance of vehicles. Huang W L et al. [7] deliberated on the task-specific performance evaluation model of Unmanned Ground Vehicles (UGVs) applicable to the annual Intelligent Vehicle Future Challenge (IVFC) competition. Wang W et al. [8] proposed a novel quantitative comprehensive performance evaluation method for autonomous vehicles, which assesses autonomous vehicles quantitatively across four aspects: driving safety, riding comfort, intelligence, and efficiency. Park J Y et al. [9] devised a control algorithm for the torque vectoring system to enhance the handling performance of green vehicles and assessed the vehicle dynamics performance, thereby boosting controllability and stability.

Node weights can be used to measure the importance of nodes. The discourse on the significance of nodes in the network model can be primarily categorized into three types: First, the assessment of the significance of network nodes [10,11]; Second, the investigation taking into account the significance of nodes [12,13]; Third, the identification of crucial nodes in the network based on the significance of nodes [14,15]. When quantifying the significance of nodes in the network model, studies can be conducted from the viewpoints of the local network [16,17], the global network [18,19], and the integration of the two [20,21]. When exploring the significance of nodes by integrating the global and local aspects of the network, Yu Jintao et al. comprehensively take into account the outgoing and incoming intensities of nodes. They quantify the local significance of nodes via the information interaction intensity of a single node, while utilizing the node efficiency to quantify the global significance of nodes.

In conclusion, the majority of prior research endeavors concentrated on the network modeling of traditional vehicles or electric vehicles. Moreover, with respect to effectiveness evaluation, it was predominantly conducted from a microscopic

viewpoint to evaluate specific vehicle performance aspects during driving, such as stability and dynamic performance. Few researchers have comprehensively considered the significance of nodes from both the global and local perspectives of the driving network for CAVs, nor have they accounted for the reliability of the driving system itself, leading to incomplete evaluation outcomes. The construction and optimization of network models are currently a research focal point, and they also represent a complex and challenging process [22,23]. The network of CAVs is a heterogeneous and open system, characterized by a complex network structure [24, 25]. Moreover, numerous aspects of this system are uncertain.

The objective of this paper is to construct a network model of the driving loop for CAVs, optimize system performance, and enhance driving safety and reliability through scientific effectiveness assessment. Based on this, this paper innovatively proposes to construct a network model for CAVs grounded in the OODA loop. Simultaneously, the integrated model of the Interpretive Structural Model (ISM) and complex network theory is employed to measure the node network weights by incorporating both global and local perspectives. Additionally, driving reliability is considered through the fuzzy evaluation method, and the node weights that account for driving reliability are comprehensively determined. Finally, the effectiveness of the CAVs driving loop network model is evaluated by enhancing the information entropy model. This approach lays a solid foundation for in-depth research on the driving process of CAVs by constructing a driving loop network system, abstracting key module nodes including perception, information, decision, execution, and target, and clearly depicting the mutual influence relationships among nodes via directed edges. In the aspect of effectiveness evaluation, by comprehensively applying the analysis of node weights and the enhanced information entropy model, considering both the global and local characteristics of the network and driving reliability, this approach enables accurate assessment of the driving loop's effectiveness, thereby facilitating the establishment and continuous optimization of the network structure of the driving model for CAVs.

## 2 Methodology

This paper initially constructs a network model of the driving loop for CAVs. Subsequently, based on the integrated model of the ISM and complex network theory, the importance of global and local nodes within the network model is respectively determined. Simultaneously, the driving reliability is analyzed through the fuzzy evaluation method, and the node weights of the network model that take driving reliability into account are comprehensively determined. Finally, the effectiveness of the CAVs' driving system is evaluated by enhancing the information entropy model.

### 2.1 Network modeling based on driving loop system

**2.1.1 Nodes modeling.** A complete driving process for vehicles resembles a cyclic structure akin to the OODA loop (Observation, Orientation, Decision, Action), driven by objectives and continuously reinforced through adjustments. The traditional driving process integrates target, perception, decision, and execution into a cyclical framework, as shown in Fig 1. Due to the advanced instant communication and intelligent information technologies utilized in the CAVs, extensive

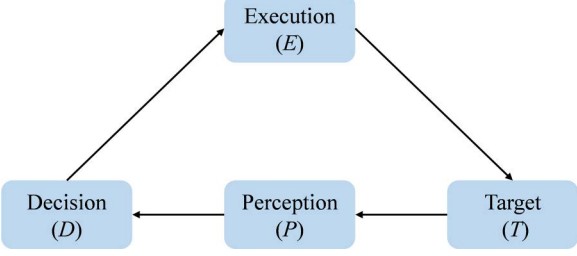

**Fig 1. Traditional vehicle driving loop network.**

information exchange is necessitated. Therefore, based on the OODA loop, the concept of a driving loop is introduced, abstracting behaviors and objectives in the driving process into nodes of Perception (P), information (I), decision (D), execution (E), and goals (T). The interrelated nodes are connected by directed edges, forming the driving loop model, as illustrated in Fig 2. If we can conduct rounds of observation and adjustment in the driving loop more efficiently and effectively, and timely coordinate all stakeholders within the driving system to make informed decisions and actions, we can enhance the intelligence and safety of the CAVs driving network. This represents one of the key objectives of this study.

In traditional modeling processes, the default evaluation of the driving system assumes a single function for the system. However, with the development of the CAVs technology, it has become necessary to account for multiple functions of the CAVs. For example, while traditional vehicles rely solely on drivers and basic sensors for scene information, the CAVs can interact with external data through intelligent cloud platforms, V2X, and other technologies. Furthermore, the CAVs decision-making is influenced by numerous factors. Focusing only on a single module result in incomplete evaluations. Therefore, a layered approach is adopted, whereby module nodes are decomposed through hierarchical mapping, as shown in Fig 3. Functional module nodes are connected through driving loop edges, and hierarchical mapping is utilized for network modeling of the driving system. Functional modules are abstracted into node sets, with multiple nodes representing each module.

**2.1.2 Edges modeling.** The transmission between driving loop nodes is directional, primarily involving six fundamental relationships: $T-P$, $P-I$, $I-D$, $D-E$, $E-T$, and others, corresponding to five types of driving loop edges.

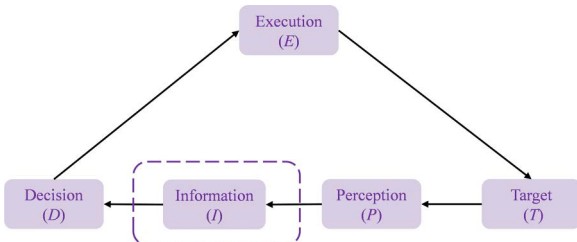

**Fig 2. The CAVs driving loop network.**

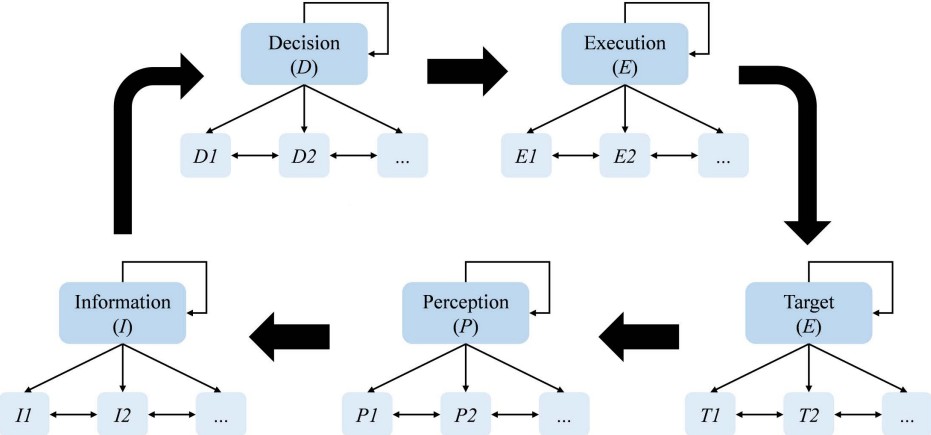

**Fig 3. Hierarchical model of driving loop.**

The effectiveness level of an edge is related to the metrics and correlations of the two nodes, which can be represented by the task demand affiliation function of the edge.

$$E_{ij} = f((x_{i1}, x_{i2}, ..., x_{in}), (x_{j1}, x_{j2}, ..., x_{jm}), O_k)$$

(1)

where $E_{ij} \in [0, 1]$ represents the edge's ability to satisfy demand. The parameters $x_{i1}, x_{i2}, ..., x_{in}$ represent the indicators associated with node $V_i$, and $x_{j1}, x_{j2}, ..., x_{jm}$ represent the indicators associated with node $V_j$. The variable $O_k$ denotes the type of relationship to which the edge belongs, $k \in \{T-P, P-I, I-P, P-D, D-P, I-D, D-E, D-T, E-T, T-E, D-D\}$.

The task demand membership function is typically derived through model simulations, data analysis, and rule-based reasoning. In practical applications, due to significant differences in functionality, technology, and real-world conditions, contributions to the overall system's driving efficiency differ significantly. Therefore, in the network model, the importance of nodes is indicated by their criticality.

## 2.2 Node weights

### 2.2.1 Node global weights based on ISM.

The Interpretive Structural Model (ISM) was proposed by Professor John N. Warfield in 1973 [26,27]. Based on graph theory, this method employs adjacency matrix principles to establish a structural model of complex systems. It decomposes the complex and disordered relationships between elements within a system into clear, hierarchical structures. ISM is widely used in fields such as key element identification and hierarchical decomposition [28,29].

The system and classification of driving loop network nodes resemble the OODA loop and follow a hierarchical structure. Therefore, this paper employs the ISM method to describe and decompose the network structure. By analyzing the interrelationships between nodes, the driving loop network can be structured into a hierarchical model, facilitating an analysis of node importance from a global network perspective.

To classify the hierarchical structure of the driving loop network, it is essential to conduct an equivalence transformation on the driving loop, dividing the target nodes into two virtual nodes: the source node $T^\alpha$ and the sink node $T^\beta$, as shown in Fig 4.

After the equivalence transformation, the equivalence adjacency matrix $A = [a_{ij}]_{N*N}$ can be expressed, where $a_{ij} = 1$ if there is a connection, or $a_{ij} = 0$ in the absence of a connection.

The steps for using the ISM method to classify the hierarchical structure of the driving loop network are as follows:

**Step 1:** Construct the equivalent adjacency matrix $A$ for the driving loop network.

**Step 2:** Calculate the reachability matrix $M$.

$$M = (A + I)^1 \neq (A + I)^2 \neq \cdots \neq (A + I)^n = (A + I)^{n+1}$$

(2)

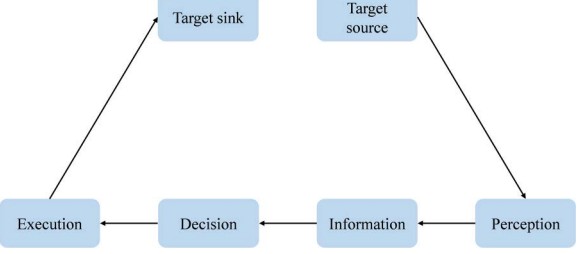

**Fig 4. Equivalent transformation model of driving loop network.**

where *I* is the identity matrix, and the calculation process must conform to the rules of the Boolean algebra.

**Step 3:** Obtain a multi-level hierarchical structure diagram through hierarchical classification.

Utilizing Equations 3 and 4, we derive the reachability set $P(V_i)$ and the predecessor set $Q(V_i)$ for node $V_i$. In this context, $P(V_i)$ denotes the set of nodes that can be influenced by $V_i$ through directed information flow, while $Q(V_i)$ represents the set of nodes capable of influencing $V_i$. Subsequently, by applying the judgment condition outlined in Equation 5, we perform node ranking to construct a multi-level directed graph.

$$P(V_i) = \{V_j | m_{ij} = 1\} \ (j = 1, 2, ..., n) \tag{3}$$

$$Q(V_i) = \{V_j | m_{ji} = 1\} \ (j = 1, 2, ..., n) \tag{4}$$

$$P(V_i) \cap Q(V_i) = P(V_i) \tag{5}$$

where $P(V_i) \cap Q(V_i)$ represents the intersection of the reachability set and the predecessor set. When this condition is met, it signifies that the nodes within this set can influence node $V_i$, but $V_i$ cannot, in turn, influence other nodes.

**Step 4:** Calculate hierarchical weights.

Through hierarchical classification, the driving loop network is structured into n levels, denoted as $L_{i'}$ ($i' = 1, 2. \cdots, n$). Different network levels correspond to distinct weights, indicating that nodes at various levels hold differing degrees of importance within the network. The network hierarchical weights serve as a crucial metric that objectively reflects these differences in importance across levels. The calculation formula is as follows:

$$\alpha_i = \frac{\frac{1}{i'}}{\sum_{i'}^n \left(\frac{1}{i'}\right)} \tag{6}$$

where $\alpha_i$ indicates the network hierarchical weights of the evaluated node $i$, and $i'$ denotes the level in which node $i$ resides.

To express the influence of the set of nodes associated with node $i$, the following formula expresses this influence:

$$\beta_i = \lambda \sum_k b_{k \to i} c_{k \to i} + (1 - \lambda) \sum_j b_{i \to j} c_{i \to j} \tag{7}$$

where $\lambda$ is the indegree node coefficient. If $\lambda > 0.5$, it suggests that the significance of the node's outdegree is inferior to that of its indegree. Here, $b_{k \to i}$ represents the network hierarchical weights of the indegree nodes associated with node $i$, $c_{k \to i}$ denotes the number of indegree nodes associated with node $i$, $b_{i \to j}$ represents the network hierarchical weights of the outdegree nodes related to node $i$, and $c_{i \to j}$ denotes the number of outdegree nodes associated with node $i$.

**Step 5:** Calculate the global network node weights.

The global weights of a node should comprehensively account for both the node's hierarchical weights and the influence of associated nodes. The following formula expresses this calculation:

$$w_i^1 = \alpha_i \cdot \beta_i \tag{8}$$

**2.2.2 Node local weights based on complex network.** Network parameters can quantitatively represent the structural characteristics of the driving loop system. For assessing the local importance of nodes, node strength is commonly utilized. However, relying solely on this single metric may lead to inaccuracies in identifying weak links within the network [30]. Therefore, this study employs both node strength and node efficiency to evaluate the local importance of nodes.

The calculation formula for node strength $k_i$ is as follows. A larger node strength indicates a higher degree of association with other nodes and greater importance within the network.

$$k_i = \lambda k_i^{in} + (1 - \lambda) k_i^{out} \tag{9}$$

where $k_i^{in}$ represents the in-degree of node $i$ and $k_i^{out}$ represents the out-degree of node $i$.

The calculation formula for node efficiency $\eta_i$ is as follows. Node efficiency indicates the average closeness of this node to other nodes; higher efficiency signifies a more important position for the node within the network.

$$\eta_i = \frac{1}{N-1} \sum_{k=1, i \neq k}^{N} \frac{1}{d_{ik}} \tag{10}$$

where $N$ is the total number of nodes in the network. If there is no path connecting node $i$ to node $k$, then $d_{ik} = \infty$ and $\eta_i = 0$.

By combining node strength and the degree of influence on other nodes, the local weights of node $i$ is given by:

$$w_i^2 = k_i \cdot \eta_i \tag{11}$$

Based on the global and local weights of the node, the overall network importance $w_i^{12}$ is expressed as:

$$w_i^{12} = w_i^1 \cdot w_i^2 \tag{12}$$

Normalizing $w_i^{12}$ yields the network weights of node $i$:

Perform normalization processing on $w_i^{12}$, and the node network weights of node $i$ can be obtained as follows:

$$w_i' = \frac{w_i^{12}}{\sum_{i=1}^{n} w_i^{12}} \tag{13}$$

**2.2.3 Analysis of Driving Reliability Based on Fuzzy Evaluation Method.** The performance of a node's function depends on the reliability of its individual functional modules. Driving reliability is defined as the extent to which a module can be reliably utilized and meet predefined requirements during the driving process. Higher driving reliability corresponds to more effective goal achievement and increases the importance of the associated network node.

Due to the complexity of the internal and external adversarial environments of the driving system, which involves numerous influencing factors, these factors are not only related to the system's own characteristics but also restricted by external interference and objective scenarios. The fuzzy comprehensive evaluation method, grounded in fuzzy mathematics, is a comprehensive evaluation approach that transforms qualitative assessment into quantitative assessment based on the theory of membership degree [31]. Therefore, the assessment of driving reliability exhibits fuzziness. Consequently, the fuzzy evaluation method is employed to assess the driving reliability. The steps for the calculation are as follows:

**Step 1:** Determine the set of evaluation factors $U$.

The set $U$ of factors affecting the reliability of driving is obtained by statistical analysis, rule inference, and expert experience for each factor in the driving loop.

$$U_i = \{x_{i1}, x_{i2}, ..., x_{in}\} \tag{14}$$

where $x_{ij}$ denotes the $j-th$ element that affects the reliability of module $i$.

**Step 2:** Determine the evaluation comment set $P_t$ and the quantitative evaluation value set $H_t$.

The evaluation results are classified into multiple levels of comments, each associated with corresponding quantitative evaluation values that range from [0,1].

**Step 3:** Determine the factor weights.

The Improved Analytic Hierarchy Process (IAHP) [32] is employed to assign weights to factors, eliminating the need for a one-time verification step and enhancing adaptability. The steps for determining weights using IAHP are as follows:

**a.** Establish the triad comparison matrix.

Perform pairwise comparisons of influencing factors at the same level, utilizing the triad scaling method for quantification, and the specific meanings are shown in Table 1.

The expression of the judgment matrix $C$ is obtained as follows.

$$C = \begin{bmatrix} C_{11} & \cdots & C_{1n} \\ \vdots & \ddots & \vdots \\ C_{n1} & \cdots & C_{nn} \end{bmatrix}$$

(15)

**b.** Construct the transfer matrix.

$$Z_{ij} = \frac{1}{n} \sum_{k=1}^{n} (c_{ik} - c_{jk})$$

(16)

**c.** Construct the proposed optimal transfer matrix $G$.

$$g_{ij} = exp\left(z_{ij}\right)$$

(17)

**d.** Determine the weight of each factor.

The eigenvalue vector of the maximum eigenvalue $\lambda_{max}$ of the matrix $G$ is calculated and normalized to obtain the weights of each factor $\xi_i' = \left(\xi_{i1}', \xi_{i2}', \ldots, \xi_{in}'\right)$.

**Step 4:** Conduct single-factor evaluation.

Utilize the results of expert evaluation, the fuzzy evaluation relation matrix $R_i$ from the factor set $U_i$ to the comment set $P_i$ is expressed as:

$$R_i = \begin{bmatrix} r_{11} & \cdots & r_{1m} \\ \vdots & \ddots & \vdots \\ r_{n1} & \cdots & r_{nm} \end{bmatrix}$$

(18)

in the formula, $r_{ij}$ represents the membership degree of the $i-th$ influencing factor belonging to the $j-th$ evaluation grade, which is obtained according to the expert evaluation.

**Table 1. Triad scaling method.**

| Comparison of the importance of $i, j$ | Take value |
|---|---|
| Same | $c_{ij} = c_{ji} = 0$ |
| $i$ stronger than $j$ | $c_{ij} = 1, c_{ji} = -1$ |

**Step 5:** Carry out comprehensive evaluation of multiple indicators.

Calculate the degree of affiliation of each rubric using module $i$ reliability.

$$T_i = R_i \cdot \xi_i'$$ 

(19)

Then transform the reliability values under external influence using the affiliation vector of module $i$.

$$w_i^3 = H_i \cdot T_i$$ 

(20)

where $H_i$ is thes value of the quantitative evaluation of the affiliation degree.

Given that a low reliability value in driving can result in increased confusion and, ultimately, decreased effectiveness, the final node weights can be derived by enhancing the traditional COX proportional hazards regression model to better connect the node's network weights and driving reliability, as follows:

$$w_i = w_i' \cdot exp\left(-w_i^3\right)$$ 

(21)

where $w_i$ represents the node weights.

The flowchart for calculating the node weights of the driving loop network model of CAVs is shown in Fig 5.

## 2.3 Effectiveness assessment of driving system based on improved information entropy

Effectiveness is fundamentally defined as the degree to which a system achieves its goals or fulfills specific task requirements. Within the context of a driving system, effectiveness is reflected in the realism and potential benefits attained in

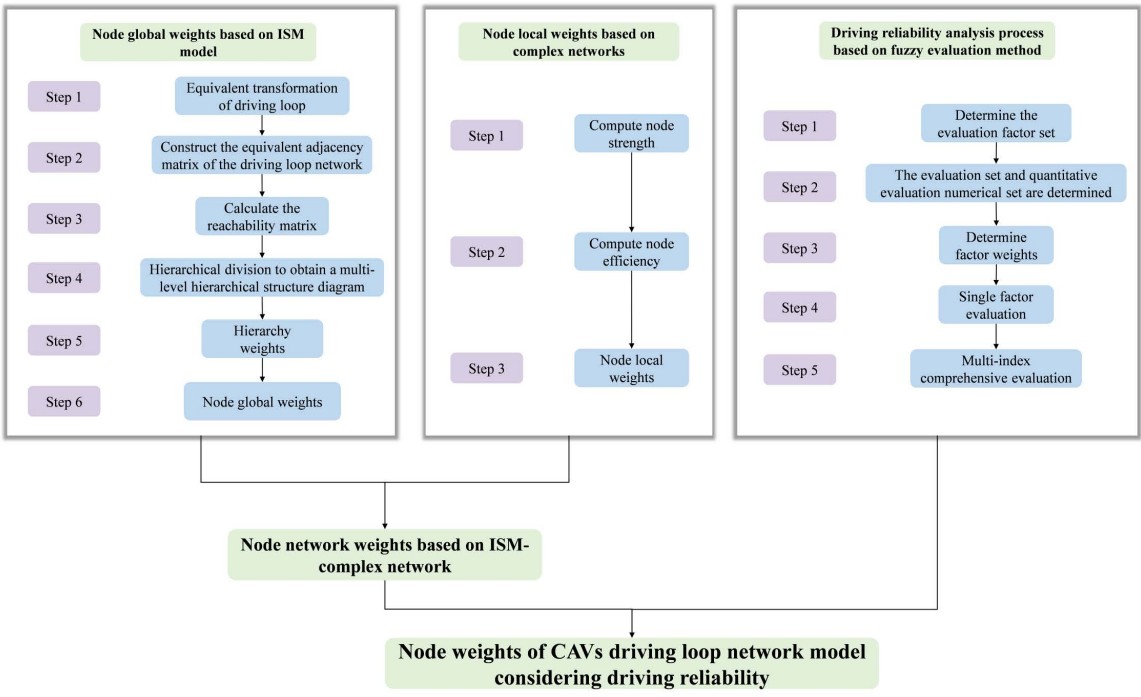

**Fig 5. Flowchart of node weights of the driving loop network model of CAVs.**

relation to driving objectives. Information entropy serves as a valuable measure of uncertainty; thus, it is utilized to evaluate the effectiveness of the driving system, considering both node networks and driving reliability [33].

According to the modeling of nodes and edges, a higher degree of task requirement membership indicates a reduction in the uncertainty associated with effectiveness, resulting in enhanced effectiveness and improved outcomes in achieving objectives. Letting the task requirement membership of an edge be denoted as $E_{ij}$ (where $0 \leq E_{ij} \leq 1$), the self-information measure $-lnE_{ij}$ represents the uncertainty associated with the effectiveness of that edge. Taking into account the impact of node weights on driving effectiveness, the overall self-information for the edge can be expressed as follows:

$$H_e = -k \cdot w_i \cdot w_j \cdot ln \, E_{ij}$$

(22)

where $w_i$ and $w_j$ are the weights of nodes $V_i$ and $V_j$, respectively; $k$ is the correction coefficient to make the comparison of the final result changes more intuitive and obvious, and the value here is $k = 10$.

The uncertain self-information of the driving loop network is the sum of the integrated self-information of each edge. The effectiveness of the traveling loop network is as follows.

$$E_{OP} = exp \, (-H_{OP})$$

(23)

For the above effectiveness formula, when the comprehensive self-information of a node is large, then the driving target effectiveness of the traveling loop will be equal to 0, which means that the information of the driving loop cannot be circulated and the desired functional goal cannot be achieved smoothly; on the contrary, if the self-information of a driving loop is 0, then the driving target effectiveness will be equal to 1 and the driving loop will complete its task more smoothly.

In the effectiveness calculation formula provided above, a very high overall self-information for a particular node result in an effectiveness of zero for the associated driving target, indicating that information cannot flow through that loop and the expected functional objectives are unachievable. Conversely, if the self-information of a driving loop is 0, the effectiveness of the associated driving target is 1, indicating that the loop will complete the task more smoothly.

When multiple loops within the driving network simultaneously contain the same target, let $x$ denote the number of driving loops containing the target $T_i$ (where $x = 2, 3 \ldots$). The uncertainty self-information for the $j-th$ driving loop is denoted as $H_{ij}$. In this context, when evaluating a single target, it is essential to assess the comprehensive self-information across multiple loops. The combined self-information for the same target is not merely the sum of the self-information from different loops. Since the completion of a target can be realized through various driving loops, the impact of the overall self-information of a single loop on that target diminishes. As the number of accessible loops increases, the total self-information for that target correspondingly decreases. Based on this reasoning, a parallel formula is utilized to calculate the uncertainty self-information for the driving target $T_i$ encompassed by multiple driving loops, expressed as follows:

$$H_i = \frac{1}{\sum\limits_{j=1}^{y} \frac{1}{H_{ij}}}$$

(24)

The effectiveness of the driving loop system for a single target is as follows.

$$E_i = exp \, (-H_i)$$

(25)

The effectiveness of the driving loop system for multiple targets is as follows.

$$E = \sum_{i=1}^{n} k_i \cdot E_i$$

(26)

where $k_i$ is the different driving target weights.

The flowchart of the effectiveness evaluation of the driving loop network system of CAVs is shown in Fig 6.

## 3 Results

### 3.1 Target-driven loop network model construction

In the target-driven loop network model, the system initiates with target $T1$. Through the perception system $P1$ and the communication and coordination system $P2$, the system perceives itself as well as other targets. Based on this perception, it collects self-information $I1$ and surrounding environment information $I2$. These interactions are then aggregated, leading to global decision-making and planning $D1$ and local decision-making and planning $D2$, which determine the optimal path $T2$. The system then executes control $E$ and, through feedback from this execution, achieves target $T1$. This process can be abstracted into a driving loop network, as illustrated in Fig 7. Table 2 specifies the symbols of each node in the network model and the modules of the CAVs driving loop that they represent.

### 3.2 Node network weights calculation

(1) Equivalent transformation of the network model and construction of the equivalent adjacency matrix.

Abstracting $T1$ and $T2$ into two nodes $T_1^\alpha$, $T_1^\beta$ and $T_2^\alpha$, $T_2^\beta$ respectively, the proposed network equivalence model is shown in Fig 8.

The equivalent adjacency matrix $A'$ is as follows.

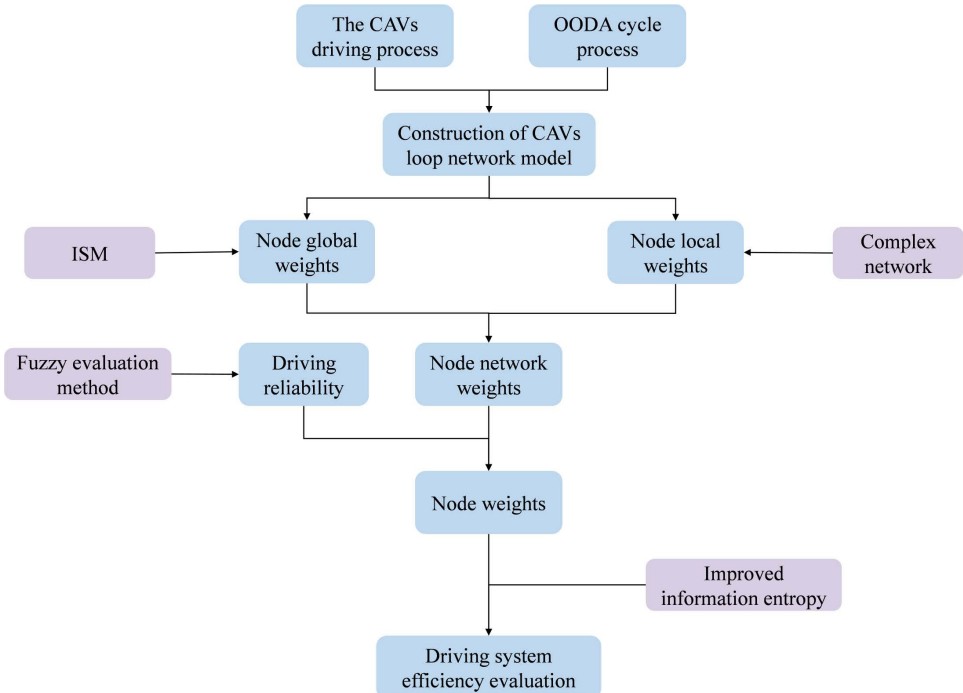

**Fig 6. Flowchart of the Effectiveness Evaluation of the Driving Loop Network System of CAVs.**

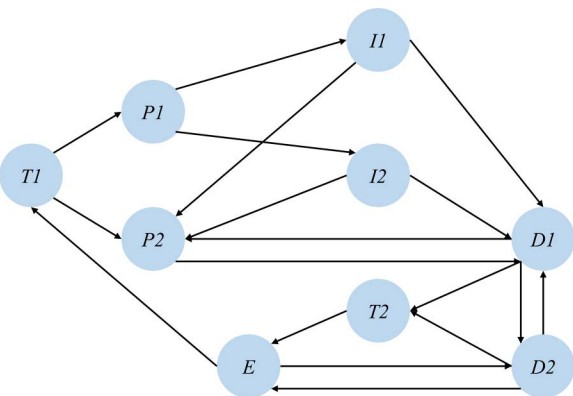

**Fig 7. Driving loop network model.**

**Table 2. Symbols of each node in the driving loop network model of CAVs and the modules they represent.**

| Modules | Node symbols | Node names |
|---|---|---|
| $T$ (Target) | $T1$ | Safe collision avoidance |
| | $T2$ | Optimal path |
| $P$ (Perception) | $P1$ | Perception system |
| | $P2$ | Communication collaboration system |
| $I$ (Information) | $I1$ | Personal information |
| | $I2$ | Surrounding environment information |
| $D$ (Decision) | $D1$ | Global decision planning |
| | $D2$ | Local decision planning |
| $E$ (Execution) | $E$ | Execution control |

$$A' = \begin{bmatrix} & T_1^{\alpha} & T_1^{\beta} & P1 & P2 & I1 & I2 & D1 & D2 & E & T_2^{\alpha} & T_2^{\beta} \\ T_1^{\alpha} & 0 & 0 & 1 & 1 & 0 & 0 & 0 & 0 & 0 & 0 & 0 \\ T_1^{\beta} & 0 & 0 & 0 & 0 & 0 & 0 & 0 & 0 & 0 & 0 & 0 \\ P1 & 0 & 0 & 0 & 0 & 1 & 1 & 0 & 0 & 0 & 0 & 0 \\ P2 & 0 & 0 & 0 & 0 & 0 & 0 & 1 & 0 & 0 & 0 & 0 \\ I1 & 0 & 0 & 0 & 1 & 0 & 0 & 1 & 0 & 0 & 0 & 0 \\ I2 & 0 & 0 & 0 & 1 & 0 & 0 & 1 & 0 & 0 & 0 & 0 \\ D1 & 0 & 0 & 0 & 1 & 0 & 0 & 0 & 1 & 0 & 1 & 0 \\ D2 & 0 & 0 & 0 & 0 & 0 & 0 & 1 & 0 & 1 & 1 & 0 \\ E & 0 & 1 & 0 & 0 & 0 & 0 & 0 & 1 & 0 & 0 & 0 \\ T_2^{\alpha} & 0 & 0 & 0 & 0 & 0 & 0 & 0 & 0 & 0 & 0 & 0 \\ T_2^{\beta} & 0 & 0 & 0 & 0 & 0 & 0 & 0 & 0 & 1 & 0 & 0 \end{bmatrix}$$

(2) The reachability matrix $M$ obtained from the rank adjacency matrix.

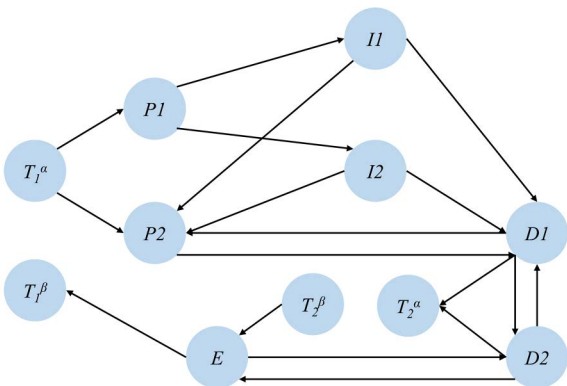

**Fig 8. Equivalence model of driving loop network.**

$$
M = \begin{bmatrix}
 & T_1^\alpha & T_1^\beta & P1 & P2 & I1 & I2 & D1 & D2 & E & T_2^\alpha & T_2^\beta \\
T_1^\alpha & 1 & 1 & 1 & 1 & 1 & 1 & 1 & 1 & 1 & 1 & 0 \\
T_1^\beta & 0 & 1 & 0 & 0 & 0 & 0 & 0 & 0 & 0 & 0 & 0 \\
P1 & 0 & 1 & 1 & 1 & 1 & 1 & 1 & 1 & 1 & 1 & 0 \\
P2 & 0 & 1 & 0 & 1 & 0 & 0 & 1 & 1 & 1 & 1 & 0 \\
I1 & 0 & 1 & 0 & 1 & 1 & 0 & 1 & 1 & 1 & 1 & 0 \\
I2 & 0 & 1 & 0 & 1 & 0 & 1 & 1 & 1 & 1 & 1 & 0 \\
D1 & 0 & 1 & 0 & 1 & 0 & 0 & 1 & 1 & 1 & 1 & 0 \\
D2 & 0 & 1 & 0 & 1 & 0 & 0 & 1 & 1 & 1 & 1 & 0 \\
E & 0 & 1 & 0 & 1 & 0 & 0 & 1 & 1 & 1 & 1 & 0 \\
T_2^\alpha & 0 & 0 & 0 & 0 & 0 & 0 & 0 & 0 & 0 & 1 & 0 \\
T_2^\beta & 0 & 1 & 0 & 1 & 0 & 0 & 1 & 1 & 1 & 1 & 1 \\
\end{bmatrix}
$$

(3) Level assignment of network nodes.

Find the reachable set $P(V_i)$ and the prior set $Q(V_i)$ of nodes and their common set $C(V_i) = P(V_i) \cap Q(V_i)$, as shown in the Table 3.

According to the hierarchy level division process in ISM, we can get this driving loop network which can be divided into five levels, i.e. $L_1 = \left\{ T_1^\beta, T_2^\alpha \right\}$, $L_2 = \{P1, D1, D2, E\}$, $L_3 = \left\{ I1, I2, E, T_2^\beta \right\}$, $L_4 = \{P1\}$, $L_5 = \{T_1^\alpha\}$。

(4) Modeling the interpretative structure of the network.

According to the level assignment of the network nodes, the ISM interpretation structure model of the corresponding network can be built, as shown in Fig 9.

As illustrated in the figure, the application of the Interpretive Structural Model (ISM) to analyze the driving loop network enables the decomposition of the original network into a more structured hierarchy, thereby clarifying the complex interrelations between nodes. The relationships between nodes across different levels in this driving loop network are both complex and interdependent. The top level of the model comprises $T_1^\beta$ and $T_2^\alpha$, which represent the ultimate goals of collision avoidance and optimal path selection. The second level, consisting of $P2$, $D1$, and $D2$, represents the direct factors influencing the final objectives and the driving loop's performance, where their effectiveness directly impacts the entire system. The third and fourth levels contain intermediate influencing factors that indirectly affect the driving loop. The

**Table 3. ISM decomposition process.**

| Node | $P(V_i)$ | $Q(V_i)$ | $C(V_i)$ |
|---|---|---|---|
| $T_1^\alpha$ | $T_1^\alpha, T_1^\beta, P1, P2, I1, I2, D1, D2, E, T_2^\alpha$ | $T_1^\alpha$ | $T_1^\alpha$ |
| $T_1^\beta$ | $T_1^\beta$ | $T_1^\alpha, T_1^\beta, P1, P2, I1, I2, D1, D2, E, T_2^\beta$ | $T_1^\beta$ |
| $P1$ | $T_1^\beta, P1, P2, I1, I2, D1, D2, E, T_2^\alpha$ | $T_1^\alpha, P1$ | $P1$ |
| $P2$ | $T_1^\beta, P2, D1, D2, E, T_2^\alpha$ | $T_1^\alpha, P1, P2, I1, I2, D1, D2, E, T_2^\beta$ | $P2, D1, D2, E$ |
| $I1$ | $T_1^\beta, P2, I1, D1, D2, E, T_2^\alpha$ | $T_1^\alpha, P1, I1$ | $I1$ |
| $I2$ | $T_1^\beta, P2, I2, D1, D2, E, T_2^\alpha$ | $T_1^\alpha, P1, I1$ | $I2$ |
| $D1$ | $T_1^\beta, P2, D1, D2, E, T_2^\alpha$ | $T_1^\alpha, P1, P2, I1, I2, D1, D2, E, T_2^\beta$ | $P2, D1, D2, E$ |
| $D2$ | $T_1^\beta, P2, D1, D2, E, T_2^\alpha$ | $T_1^\alpha, P1, P2, I1, I2, D1, D2, E, T_2^\beta$ | $P2, D1, D2, E$ |
| $E$ | $T_1^\beta, P2, D1, D2, E, T_2^\alpha$ | $T_1^\alpha, P1, P2, I1, I2, D1, D2, E, T_2^\beta$ | $P2, D1, D2, E$ |
| $T_2^\alpha$ | $T_2^\alpha$ | $T_1^\alpha, P1, P2, I1, I2, D1, D2, E, T_2^\alpha, T_2^\beta$ | $T_2^\alpha$ |
| $T_2^\beta$ | $T_1^\beta, P2, D1, D2, E, T_2^\alpha, T_2^\beta$ | $T_2^\beta$ | $T_2^\beta$ |

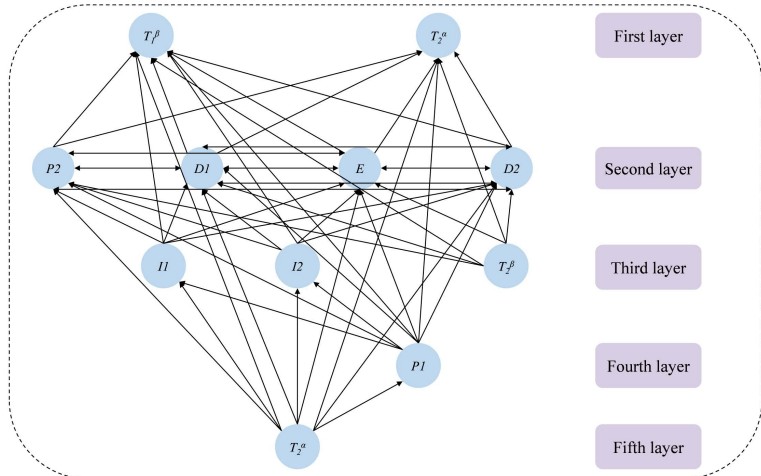

**Fig 9. ISM model of driving loop network.**

fifth level represents the fundamental root factors, where $T_1^\alpha$ serves as the driving goal for the entire driving loop network system.

(5) Calculation of global weights of node networks.

According to the formula, the weights of each level can be obtained:

$$\alpha = (0.4380, 0.2190, 0.1460, 0.1095, 0.0875)$$

Take the in-degree node coefficient $\lambda = 0.6$, and the degree of influence of the nodes associated with the node on the node is obtained according to Equation 7, and then the global weights of the network of the node is obtained, as shown in the following Table 4.

(6) Calculation of local weights of node networks.

According to Equations 9 and 10, the node strength and node efficiency corresponding to the nodes can be obtained from the local weights of the nodes. Then the global weights obtained in Equation 7, the network importance and network weights of the nodes are finally obtained, as shown in the following Table 4.

Since node $T_1^\alpha$ and node $T_2^\alpha$ are the final target sinks, they cannot contribute to other nodes from the network structure level, so the final network weight obtained is 0.

### 3.3 Driving reliability analysis and node weights calculation

Taking node $P1$ as an example, the reliability of this node is calculated using the fuzzy evaluation method, and the calculation process is as follows.

(1) The set of factors affecting the reliability of node $P1$ is determined as $U = \left\{ \begin{array}{c} x_1 \\ x_2 \\ x_3 \end{array} \right\}^T = \left\{ \begin{array}{c} Accuracy \\ Environmental adaptability \\ Comprehensiveness \end{array} \right.$

(2) Determine the set of fuzzy rubrics and the fuzzy evaluation value: $P = \left\{ \begin{array}{c} High reliability \\ Higher reliability \\ Average reliability \\ Poor reliability \\ Very poor reliability \end{array} \right.$

(3) Determination of factor weights:

The judgment matrix is obtained using the three-scale method: $C = \begin{array}{c} x_1 \\ x_2 \\ x_3 \end{array} \left\{ \begin{array}{ccc} 0 & 1 & 1 \\ -1 & 0 & -1 \\ -1 & 1 & 0 \end{array} \right\}$. Then, according to the

formula, the transfer matrix and the optimization transfer matrix are obtained, and finally the weights of the evaluation factors $\xi'$ are calculated: $\xi' = (0.5627, 0.2889, 0.1484)$;

Affiliation degree fuzzy evaluation matrix: $R = \begin{bmatrix} 0.6 & 0.3 & 0.1 & 0 & 0 \\ 0.5 & 0.2 & 0.3 & 0 & 0 \\ 0.5 & 0.4 & 0.1 & 0 & 0 \end{bmatrix}$; the multi-indicator evaluation:

$T = \xi' * R = [0.5563, 0.2860, 0.1578, 0, 0]$; in summary, the final node reliability value about node $P1$ is $w_i^3 = T \cdot H^T = 0.8497$.

Similarly, the reliability values and the final node weights of all nodes in the driving loop system can be obtained, as shown in Table 5 below.

**Table 4. Statistics of node weights.**

| Nodes | $\alpha$ | $\beta$ | $k_i$ | $\eta_i$ | $w_i^1$ | $w_i^2$ | $w_i^{12}$ | $w_i'$ |
|---|---|---|---|---|---|---|---|---|
| $T_1^\alpha$ | 0.0875 | 0.9498 | 3.8 | 0.1617 | 0.0831 | 0.6145 | 0.0511 | 0.0122 |
| $T_1^\beta$ | 0.4380 | 0.2049 | 2.2 | 0 | 0.0897 | 0 | 0 | 0 |
| $P1$ | 0.1095 | 0.9796 | 4 | 0.4117 | 0.1073 | 1.6468 | 0.1766 | 0.0423 |
| $P2$ | 0.2190 | 1.6074 | 7 | 0.2583 | 0.3520 | 1.8081 | 0.6365 | 0.1524 |
| $I1$ | 0.1460 | 0.965 | 3.8 | 0.3583 | 0.1409 | 1.3615 | 0.1918 | 0.0459 |
| $I2$ | 0.1460 | 0.965 | 3.8 | 0.3583 | 0.1409 | 1.3615 | 0.1918 | 0.0459 |
| $D1$ | 0.2190 | 1.6074 | 7 | 0.3833 | 0.3520 | 2.6831 | 0.9445 | 0.2261 |
| $D2$ | 0.2190 | 1.6074 | 7 | 0.4 | 0.3520 | 2.800 | 0.9857 | 0.236 |
| $E$ | 0.2190 | 1.6074 | 7 | 0.3333 | 0.3520 | 2.3331 | 0.8213 | 0.1966 |
| $T_2^\alpha$ | 0.4380 | 1.3446 | 6.4 | 0 | 0.5889 | 0 | 0 | 0 |
| $T_2^\beta$ | 0.1460 | 1.2264 | 3.4 | 0.2917 | 0.1791 | 0.9918 | 0.1776 | 0.0426 |

**Table 5. Node reliability values and node weights.**

| Nodes | $w_i^1$ | $w_i$ |
|---|---|---|
| $P1$ | 0.8497 | 0.0181 |
| $P2$ | 0.8722 | 0.0637 |
| $I1$ | 0.9354 | 0.0180 |
| $I2$ | 0.9002 | 0.0187 |
| $D1$ | 0.8930 | 0.0926 |
| $D2$ | 0.9071 | 0.0953 |
| $T1$ | 0.8914 | 0.0050 |
| $T2$ | 0.9058 | 0.0172 |
| $E$ | 0.9739 | 0.0742 |

## 3.4 System effectiveness analysis based on information entropy

For the task demand affiliation $E$ of node-to-node edges, which is obtained according to the expert judgment and the above-mentioned fuzzy evaluation method, the integrated information quantity of each edge in the travel loop network system is then calculated as shown in Table 6 below.

With the driving loop network model shown in Fig 6, the number of driving loops containing the target nodes $T1$ and $T2$ can be obtained as $N = (T1) = 7$ and $N = (T2) = 2$, respectively. The uncertainty self-information of the driving loops is obtained by combining the integrated self-information of each edge obtained above, as shown in Table 7.

As presented in Table 8, the collision avoidance effectiveness in the driving loop network model is 0.99584, and the optimal path effectiveness is 0.99215. Whether considering a single-objective or multi-objective driving loop, and regardless of the distribution of multi-objective weights, the overall driving efficiency of the system exceeds 0.99215. This demonstrates that the driving loop network system exhibits a high expected value in achieving system goals, offering substantial actual and potential benefits, confirming the network model's reliability.

## 3.5 Model verification and comparative analysis

To further validate the methods used in this study, based on the constructed driving loop network model and performance evaluation, several scenarios were adjusted for a comparative analysis. The corresponding number of driving loops and the efficiency of the network system under different conditions were calculated, as shown in Table 9.

**Table 6. Amount of self-information at the edge of the driving loop network.**

| Name | $T1-P1$ | $T1-P2$ | $P1-I1$ | $P1-I2$ | $I1-P2$ | $E-D2$ |
|---|---|---|---|---|---|---|
| $E = f(v_i, v_j)$ | 0.68 | 0.61 | 0.94 | 0.85 | 0.82 | 0.85 |
| self-information | 0.3857 | 0.4943 | 0.0619 | 0.1625 | 0.1985 | 0.1625 |
| Comprehensive self-information | 0.0003 | 0.0016 | 0.0002 | 0.0005 | 0.0023 | 0.0115 |
| Name | $I1-P2$ | $P2-D1$ | $I1-D1$ | $I2-D1$ | $D1-P2$ | $E-T1$ |
| $E = f(v_i, v_j)$ | 0.83 | 0.89 | 0.97 | 0.91 | 0.77 | 0.78 |
| self-information | 0.1863 | 0.1165 | 0.0305 | 0.0943 | 0.2614 | 0.2485 |
| Comprehensive self-information | 0.0022 | 0.0069 | 0.0005 | 0.0016 | 0.0154 | 0.0009 |
| Name | $D1-D2$ | $D1-T2$ | $D2-T2$ | $D2-D1$ | $T2-E$ | $D2-E$ |
| $E = f(v_i, v_j)$ | 0.91 | 0.98 | 0.96 | 0.89 | 0.99 | 0.86 |
| self-information | 0.0943 | 0.0202 | 0.0408 | 0.1165 | 0.0101 | 0.1508 |
| Comprehensive self-information | 0.0083 | 0.0003 | 0.0007 | 0.0100 | 0.0001 | 0.0107 |

**Table 7. Target node driving loops and their uncertain self-information quantity.**

| Target | Number | Driving loop | Uncertainty self-information |
|---|---|---|---|
| $T1$ | 1 | $T1 - P1 - I1 - D1 - D2 - E - T1$ | 0.0209 |
| | 2 | $T1 - P1 - I2 - D1 - D2 - E - T1$ | 0.0223 |
| | 3 | $T1 - P1 - I1 - P2 - D1 - D2 - E - T1$ | 0.0296 |
| | 4 | $T1 - P2 - D1 - D2 - E - T1$ | 0.0284 |
| | 5 | $T1 - P1 - I1 - D1 - P2 - D1 - D2 - E - T1$ | 0.0432 |
| | 6 | $T1 - P1 - I2 - D1 - P2 - D1 - D2 - E - T1$ | 0.0446 |
| | 7 | $T1 - P1 - I2 - P2 - D1 - D2 - E - T1$ | 0.0308 |
| $T2$ | 1 | $T2 - E - D2 - T2$ | 0.0123 |
| | 2 | $T2 - E - D2 - D1 - T2$ | 0.0219 |

**Table 8. The effectiveness of the driving loop network at the target node.**

| Driving target | $H_i$ | $E_i$ |
|---|---|---|
| $T1$ | 0.00417 | 0.99584 |
| $T2$ | 0.00788 | 0.99215 |

**Table 9. Number of driving loops and system effectiveness in each case.**

| | Scenario 1 | Scenario 2 | Scenario 3 |
|---|---|---|---|
| Number of driving loops | 1 | 1 | 1/1 |
| System effectiveness | 0.87634 | 0.89808 | 0.89521/0.89395 |
| Number of targets | 1 | 1 | 2 |
| | Scenario 4 | Scenario 5 | This paper scenario |
| Number of driving loops | 2 | 2/3 | 7/2 |
| System effectiveness | 0.90069/0.90547 | 0.97099/0.97318 | 0.99584/0.99215 |
| Number of targets | 1 | 2 | 3 |

Scenario 1: Only the traditional driving loop network is considered, consisting of target, perception, decision, and execution cycles, without external information interaction and with only a single objective.

Scenario 2: On the basis of Scenario 1, the driving reliability is not taken into account, that is, the driving reliability equals 1.

Scenario 3: Based on Scenario 1, consider multiple objectives, that is, multiple $T$ nodes.

Scenario 4: Based on Scenario 1, the system is further subdivided, but the information feedback adjustment between nodes is not considered, that is, there are multiple $P$ nodes.

Scenario 5: Based on Scenario 4, consider adding objectives, that is, multiple $T$ and $P$ nodes.

This paper scenario: Based on Situation 5, consider the information interaction, that is, add the information set $I$ module.

Several observations can be drawn from analyzing the modeling process and Table 9. When all other conditions are constant, disregarding reliability (assuming it equals 1) leads to an overestimation of the final performance evaluation, producing less credible results (comparison between Scenario 1 and Scenario 2). The lower system efficiency observed in Scenario 1 may stem from ambiguous connections between task nodes, which result in a low degree of task affiliation

among the nodes. An increase in the number of objectives raises both the number of nodes in the network and the system efficiency (comparison between Scenario 1 and Scenario 3, and Scenario 4 and Scenario 5).

If certain nodes possess excessively high network weights, the overall efficiency decreases. This occurs because when one node's importance significantly outweighs that of others, network complexity rises, increasing vulnerability to system failure if that node experiences issues. As the number of driving loops increases, a single objective can be achieved through multiple loops, reducing the impact of any single loop's uncertainty on the overall objective. This also boosts overall efficiency, as more available loops facilitate the objective's completion. The driving loop efficiency of the CAVs, which incorporate information interaction, exceeds that of traditional vehicles (comparison between Scenario 5 and the current study).

## 4 Discussion

From Table 4 and Table 5, it is evident that the network weights and node weights of $P_2$ (Communication coordination system), $D_1$ (Global decision planning), $D_2$ (Local decision planning), and $E$ (Execution control) are higher than those of other nodes. This result supports the critical role of information acquisition for decision-making, planning, and execution within the driving process of the CAVs, indicating the feasibility of using ISM and complex networks to evaluate the importance of driving loop nodes. In practical terms, this process can be articulated as follows: first, the CAVs receives information from both itself and the external environment; next, the decision system engages in competition and coordination among itself and other entities in the environment; finally, it performs execution control and transmits execution information to other entities on the road for necessary adjustments, thereby perpetuating this cycle. If the weights of these nodes are low and their reliability is insufficient, any error in one of the steps can lead to compounded errors throughout the cyclic process, potentially resulting in more severe accidents in the vicinity. Conversely, when the entire driving loop exhibits high efficiency, the correct operations and adjustments across the system will align more closely with the established objectives of the driving loop.

When considering the efficiency of the driving loop network system, a node efficiency of zero indicates that it cannot contribute to the network. If the reliability of a node is low, it results in decreased node efficiency, ultimately diminishing the overall efficiency of the driving loop network. Consequently, the importance of that node will diminish. Therefore, coupling network weights with node driving reliability is a feasible approach.

Although this study contributes to the construction of network models and the evaluation and optimization of efficiency for the CAVs, several aspects of our work require improvement in the future. In the evaluation process, the task demand degree of the edges between various functional nodes cannot be directly obtained; consequently, we simplified the simulation and computation of this degree by relying on traditional fuzzy judgment. Furthermore, the driving process may be affected by sudden external factors. For instance, while the vehicle may detect dangers early, road and traffic conditions may leave no available space to avoid a collision. In such instances, the vehicle reliability assessed in this study may be overestimated relative to actual conditions. Additionally, since the CAVs systems and architectures possess learning capabilities, certain aspects may improve over time, necessitating ongoing research and modifications to the methods presented in this paper.

## 5 Conclusion

This paper investigates network model construction and driving loop efficiency evaluation for the CAVs by proposing a method for network modeling and efficiency assessment. This method comprehensively considers the various module nodes in the driving process, their interrelationships, and their reliability during operation. It employs node importance and driving reliability analyses to derive node weights and utilizes an improved information entropy model to evaluate the efficiency of the CAVs driving process. Additionally, a target-driven driving loop network model was developed, along with several comparative network models for analysis and discussion. The results indicate that the proposed method is

feasible, effectively models the driving network, and assesses its efficiency in alignment with subjective judgments. Furthermore, it illustrates the superiority of the CAVs over traditional vehicles and contributes to the establishment and continuous optimization of the CAVs driving model network structure. This method can also be refined further by subdividing the functions of each module, enhancing the precision of driving network model construction, and improving simulations and computations. Future improvements will consider these aspects, thereby making the final efficiency assessment more persuasive.

## Supporting information

**S1. Data 1.**
(XLSX)

**S2. Data 2.**
(XLSX)

**S3. Data 3.**
(XLSX)

**S4. Data 4.**
(XLSX)

## Acknowledgments

Thanks to the lab team for their efforts.

## Author contributions

Conceptualization: Daowen Zhang.

Data curation: Junlian Yan.

Funding acquisition: Yihong Zhang.

Investigation: Jixiang Yang, Lei Xu.

Methodology: Daowen Zhang.

Software: Junlian Yan, Qirui Luo.

Supervision: Daowen Zhang.

Validation: Jixiang Yang, Lei Xu.

Visualization: Hao Xu.

Writing – original draft: Junlian Yan.

Writing – review & editing: Qirui Luo.

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
