## [Decision Letter · Decision Letter 0]

PONE-D-25-02537Network Model Construction and Driving-Loop Performance Evaluation for Connected and Automated VehiclesPLOS ONE

Dear Dr. Zhang,

Thank you for submitting your manuscript to PLOS ONE. After careful consideration, we feel that it has merit but does not fully meet PLOS ONE’s publication criteria as it currently stands. Therefore, we invite you to submit a revised version of the manuscript that addresses the points raised during the review process.

**ACADEMIC EDITOR: please revise**==============================

We look forward to receiving your revised manuscript.

Kind regards,

Zhengmao Li

Academic Editor

PLOS ONE

Journal Requirements:

3. Thank you for stating the following financial disclosure: Project of National Automobile Accident In-depth Investigation System 202248 

Research on the key technologies of in-depth investigation and accident reconstruction of intelligent vehicle 282023Y-10408/2023MK185 

Research on Scene Reconstruction of Traffic Accidents Involving Intelligent Vehicles and Its Application in Safety Assessment KF202211

Additional Editor Comments:

Given the comments, last chance to revise

Reviewers' comments:

Reviewer's Responses to Questions

**Comments to the Author**

1. Is the manuscript technically sound, and do the data support the conclusions?

Reviewer #1: Partly

Reviewer #2: Yes

2. Has the statistical analysis been performed appropriately and rigorously? 

Reviewer #1: No

Reviewer #2: Yes

3. Have the authors made all data underlying the findings in their manuscript fully available?

Reviewer #1: No

Reviewer #2: Yes

4. Is the manuscript presented in an intelligible fashion and written in standard English?

Reviewer #1: No

Reviewer #2: Yes

5. Review Comments to the Author

Reviewer #1: The work conducts research on the development, evaluation, and continuous optimization of the driving model network structure for the CAVs. The following are some concerns and suggestions that might help to improve the work.

1. The research lacks focus, and the paper's organization is described as messy. As a result, the contributions are not clear.

2. The abstract should be revised to better highlight the novelty of the work. Clearly articulating the innovative aspects of this study will help engage readers and set the stage for the detailed discussion that follows.

3. The quality of the figures is low and the images are unclear. Enhancing the clarity, resolution, and overall presentation of the figures will make them more effective in supporting and illustrating the manuscript's content.

4. It is suggested to use figures to visualize the steps involved in using the ISM method and the fuzzy evaluation method. Visual aids like these can greatly enhance understanding and accessibility of the methodologies employed.

5. The description of the data and settings for the case study is missing or unclear.

6. The manuscript lacks a quantitative comparison with alternative modeling or evaluation methods.

7. The number of recent references, from the last two years, is limited. Including more up-to-date references will ensure that the research is grounded in the current state of knowledge and reflects the latest developments in the field.

Reviewer #2: This paper focuses on the construction and performance evaluation of the driving loop network model for connected and autonomous vehicles (CAVs). It innovatively introduces the OODA loop theory, combines the node importance with the improved information entropy model for efficiency analysis, and the method has theoretical value. My comments are as follows:

The derivation processes of the node weight calculation (such as Formulas 7-13) and the improvement of the information entropy model (Formulas 22-26) lack detailed explanations, and the basis for the values of key parameters (such as the values of λ and k) is not clearly stated. It is recommended to supplement the mathematical derivations and parameter sensitivity analysis to ensure the reproducibility of the method.

For the key data such as the "membership degree of task requirements" in Tables 5-8, it only mentions the "fuzzy statistical method", without disclosing the sources of the specific dataset, the collection conditions, and the processing procedures.

The division criteria for different scenarios in Table 8 (such as "whether reliability is considered") are not clearly defined, and the descriptions of the differences in some scenarios (such as Scenario 5 and the solution in this paper) are ambiguous.

The study only verifies the method based on a single-objective-driven model (Figures 5-7), and does not involve the robustness tests under complex road conditions (such as sudden obstacles and multi-vehicle coordination).

The analysis of the current situation of the research on the CAVs network model in the introduction section is rather general, and it does not deeply compare the advantages and disadvantages between the existing methods (such as reinforcement learning and game theory models) and the ISM method in this paper. It is necessary to expand the literature review and highlight the irreplaceability of the innovation points of this paper.

6. PLOS authors have the option to publish the peer review history of their article (what does this mean? ). If published, this will include your full peer review and any attached files.

**Do you want your identity to be public for this peer review?** For information about this choice, including consent withdrawal, please see our Privacy Policy .

Reviewer #1: No

Reviewer #2: No

---

## [Author Response · Author response to Decision Letter 1]

14 May 2025

Dear editors and reviewers

We are very grateful for your constructive comments and suggestions for our manuscript entitled "Effectiveness Evaluation of Connected and Automated Vehicles' Driving Loop: Node Weights and Driving Reliability" (Manuscript Number: PONE-D-25-02537). Your comments are very valuable and helpful for improving our manuscript. We have carefully revised the manuscript and provided point-by-point response below.

Reviewer 1

Comments 1: The research lacks focus, and the paper's organization is described as messy. As a result, the contributions are not clear.

Response 1: Thank you very much for your valuable suggestions. Regarding the issues you mentioned, we have redrafted the title and modified it to "Effectiveness Evaluation of Connected and Automated Vehicles' Driving Loop: Node Weights and Driving Reliability". At the same time, we have also reorganized the introduction part, elaborated in detail on the research progress of vehicle network modeling, vehicle efficiency evaluation, and node importance, and deeply explored the deficiencies of existing methods. In the "Contributions of This Paper" section, we have rewritten the content and clearly expounded the research objectives and innovative points. In the methods section, we have modified the titles of some chapters, taking the paper title "Effectiveness Evaluation of Connected and Automated Vehicles' Driving Loop: Node Weights and Driving Reliability" as the main line. Meanwhile, we have added a flowchart for calculating the node weights and a flowchart of the entire research process. The former presents in detail the specific steps for calculating the node weights by using the Interpretive Structural Modeling (ISM), complex network theory, and taking driving reliability into account. The latter clearly demonstrates the whole process from the construction of the network model, to the analysis of node importance using various methods, the evaluation of driving reliability, and finally to the completion of the efficiency evaluation of the network model through the improved information entropy method.

We sincerely hope that these revisions can make the structure of the paper clearer and the logic more rigorous, thus facilitating readers' understanding. The following are the revised paragraphs and the added visual graphics:

1 Introduction

With the continuous advancement of technology, including the rapid development of sensor technology, communication technology, and artificial intelligence, the Connected and Automated Vehicles (CAVs) have emerged as a crucial direction for the transformation and upgrading of the automotive industry. Additionally, the effectiveness evaluation of CAVs during the driving process has emerged as one of the prominent topics in automotive research. Vehicle effectiveness evaluation is a process that comprehensively and systematically analyzes and assesses a vehicle's capability, efficiency in accomplishing various tasks, and the extent to which goals are achieved in specific usage scenarios. With the continuous evolution of systems science and complex network theory, their applications in effectiveness evaluation are becoming more and more extensive.

In previous research on vehicle network modeling, Yang L et al. [1] proposed a vehicle network model based on the peer-to-peer (P2P) network, aiming to enhance the network's fault tolerance and ensure the stability of the network system. Ma J et al. [2] developed a stochastic electric vehicle network model that incorporated environmental costs and proposed an algorithm for solving this network model. Tu Q et al. [3] proposed a reliability-based equilibrium model for electric vehicle networks, which was employed to predict the traffic flow patterns in the road networks of both electric vehicles and gasoline vehicles. Xu Q et al. [4] constructed and validated a hybrid traffic network model in which human-driven vehicles and CAVs coexisted.

Regarding the research on vehicle effectiveness, Zhao J et al. [5] established an overall subjective evaluation model associated with three objective indicators, leveraging the Probabilistic Neural Network (PNN) to assess the dynamic performance of vehicles. Researchers such as Wang B [6] endeavored to utilize the characteristic indicators of wheels to assess the stability performance of vehicles. Huang W L et al. [7] deliberated on the task-specific performance evaluation model of Unmanned Ground Vehicles (UGVs) applicable to the annual Intelligent Vehicle Future Challenge (IVFC) competition. Wang W et al. [8] proposed a novel quantitative comprehensive performance evaluation method for autonomous vehicles, which assesses autonomous vehicles quantitatively across four aspects: driving safety, riding comfort, intelligence, and efficiency. Park J Y et al. [9] devised a control algorithm for the torque vectoring system to enhance the handling performance of green vehicles and assessed the vehicle dynamics performance, thereby boosting controllability and stability.

Node weights can be used to measure the importance of nodes. The discourse on the significance of nodes in the network model can be primarily categorized into three types: First, the assessment of the significance of network nodes [10,11]; Second, the investigation taking into account the significance of nodes [12,13]; Third, the identification of crucial nodes in the network based on the significance of nodes [14,15]. When quantifying the significance of nodes in the network model, studies can be conducted from the viewpoints of the local network [16,17], the global network [18,19], and the integration of the two [20,21]. When exploring the significance of nodes by integrating the global and local aspects of the network, Yu Jintao et al. comprehensively take into account the outgoing and incoming intensities of nodes. They quantify the local significance of nodes via the information interaction intensity of a single node, while utilizing the node efficiency to quantify the global significance of nodes.

In conclusion, the majority of prior research endeavors concentrated on the network modeling of traditional vehicles or electric vehicles. Moreover, with respect to effectiveness evaluation, it was predominantly conducted from a microscopic viewpoint to evaluate specific vehicle performance aspects during driving, such as stability and dynamic performance. Few researchers have comprehensively considered the significance of nodes from both the global and local perspectives of the driving network for CAVs, nor have they accounted for the reliability of the driving system itself, leading to incomplete evaluation outcomes. The construction and optimization of network models are currently a research focal point, and they also represent a complex and challenging process [22,23]. The network of CAVs is a heterogeneous and open system, characterized by a complex network structure [24,25]. Moreover, numerous aspects of this system are uncertain.

The objective of this paper is to construct a network model of the driving loop for CAVs, optimize system performance, and enhance driving safety and reliability through scientific effectiveness assessment. Based on this, this paper innovatively proposes to construct a network model for CAVs grounded in the OODA loop. Simultaneously, the integrated model of the Interpretive Structural Model (ISM) and complex network theory is employed to measure the node network weights by incorporating both global and local perspectives. Additionally, driving reliability is considered through the fuzzy evaluation method, and the node weights that account for driving reliability are comprehensively determined. Finally, the effectiveness of the CAVs driving loop network model is evaluated by enhancing the information entropy model. This approach lays a solid foundation for in-depth research on the driving process of CAVs by constructing a driving loop network system, abstracting key module nodes including perception, information, decision, execution, and target, and clearly depicting the mutual influence relationships among nodes via directed edges. In the aspect of effectiveness evaluation, by comprehensively applying the analysis of node weights and the enhanced information entropy model, considering both the global and local characteristics of the network and driving reliability, this approach enables accurate assessment of the driving loop's effectiveness, thereby facilitating the establishment and continuous optimization of the network structure of the driving model for CAVs.

The added flowchart:

Fig. 5. Flowchart of Node Weights of the Driving Loop Network Model of CAVs.

Fig. 6. Flowchart of the Effectiveness Evaluation of the Driving Loop Network System of CAVs.

Comments 2: The abstract should be revised to better highlight the novelty of the work. Clearly articulating the innovative aspects of this study will help engage readers and set the stage for the detailed discussion that follows.

Response 2: Thank you very much for your valuable comments on the abstract of the paper. We fully agree with the view that the abstract should better highlight the innovation of the work. After carefully studying your feedback, we immediately revised the abstract. The following is the revised abstract.

Abstract: As an emerging development trend in the automotive industry, the construction of the network model and the effectiveness evaluation of the driving loop for Connected and Automated vehicles (CAVs) are of significant importance. The objective of this paper is to construct a network model of the driving loop for CAVs and evaluate the effectiveness of the model, thereby optimizing system performance and enhancing driving safety and reliability. In this study, by integrating the driving process of CAVs and introducing the concept of the Observation, Orientation, Decision, and Action (OODA) loop, a network model of the driving loop for CAVs was established, enabling effective modeling of the complex driving process. For effectiveness evaluation, a method is proposed. This method measures the importance of nodes using the Interpretive Structural Model (ISM) and complex network theory, considers driving reliability through the fuzzy evaluation method, and comprehensively determines the node weights of the network model. Subsequently, by utilizing the node weights to enhance the information entropy model, a scientific evaluation of the CAVs' driving loop effectiveness is achieved. Through comparisons and validations across several scenarios, it has been demonstrated that this method can be effectively applied to the planning, modeling, evaluation, and optimization of CAVs network models.

Comments 3: The quality of the figures is low and the images are unclear. Enhancing the clarity, resolution, and overall presentation of the figures will make them more effective in supporting and illustrating the manuscript's content.

Response 3: Thank you for your kind suggestion. In response, we have optimized the clarity, resolution and overall presentation of the charts.

Comments 4: It is suggested to use figures to visualize the steps involved in using the ISM method and the fuzzy evaluation method. Visual aids like these can greatly enhance understanding and accessibility of the methodologies employed.

Response 4: We are extremely grateful to you for pointing out the deficiencies in the visualization of the methods in the paper. Upon receiving your feedback, we promptly carried out optimization of this part, aiming to comprehensively enhance the clarity and intuitiveness of the method description.

We have drawn a detailed flowchart covering the entire research process, which is a schematic diagram of the overall process from the construction of the CAVs network model to the final efficiency evaluation. This diagram links together all the key stages from the construction of the network model of CAVs to the efficiency evaluation of the driving system. Each link is presented as a clear module in the diagram, and arrows are used to indicate the flow direction, supplemented by necessary textual annotations, making the research process clear at a glance.

In addition, for the key link of calculating the node weights, we have specially drawn a flowchart to meticulously display the implementation steps of methods such as Interpretive Structural Modeling (ISM), complex network theory, and driving reliability. The logical relationships between each step are clear and obvious, allowing readers to quickly grasp the principles and processes of calculating the node weights.

Through this series of flowcharts, not only can readers easily understand the research methods of the paper, but also the logic and professionalism of the paper have been further enhanced. Once again, thank you for your valuable suggestions, and we hope that the revised paper will meet with your approval.

Comments 5: The description of the data and settings for the case study is missing or unclear.

Response 5: Thank you for pointing out the problems in the description of data and settings in the case study of the paper. Here is a detailed response to your questions: The data in this paper does not come from external existing datasets, but is gradually generated during the research process of the paper.

At the beginning of the research, we deeply analyzed the driving process of CAVs, combined it with the cyclic process of the Observation, Orientation, Decision, and Action (OODA) loop, and constructed the driving loop network model of CAVs. At this stage, based on the actual perception, decision, execution and other links of CAVs, we defined perception-type, information-type, decision-making-type, execution-type and target-type nodes, and used directed edges to represent the mutual influence relationships between nodes. The data was generated during the process of determining the node types and the connection relationships of the edges.

Subsequently, in order to analyze the node weights of the driving loop network model of CAVs, we applied the combined model of Interpretive Structural Modeling (ISM) and complex networks. In this step, based on the adjacency matrix, we generated the reachability matrix, divided the network hierarchy, and then determined the hierarchical weights of each node to obtain the global weights of the nodes. At the same time, with the help of complex network theory, we calculated parameters such as node strength and efficiency to measure the importance of nodes from a local perspective and obtained the local weights of the nodes. During this period, the data was generated during the processes of matrix calculation, parameter solving and model operation. When analyzing the driving reliability of CAVs based on the fuzzy evaluation method, by determining the evaluation factor set, the evaluation set and using the improved Analytic Hierarchy Process (AHP) to determine the factor weights, we obtained the driving reliability data, and integrated it with the node importance data to obtain the node weight data of the driving loop network model of CAVs considering driving reliability.

Finally, we adopted the method of improved information entropy to evaluate the efficiency of the entire driving loop network model of CAVs. In this process, combined with the obtained node weights and driving reliability data, we calculated the information entropy of each driving loop, and then obtained the overall efficiency data of the network model.

Comments 6: The manuscript lacks a quantitative comparison with alternative modeling or evaluation methods.

Response 6: Thank you very much for pointing out the lack of quantitative comparison with alternative modeling or evaluation methods in our paper. This is indeed a current shortcoming of our research. Due to time and resource constraints at this stage, we are unable to complete the supplementation of this part in the current revision. However, we have formulated a detailed follow-up research plan and will improve this section in the future. In the upcoming research, we plan to select a variety of alternative methods widely used in the field of CAVs, such as reinforcement learning, game theory models, and deep learning-related models, etc., to conduct a comprehensive quantitati

---

## [Decision Letter · Decision Letter 1]

Effectiveness Evaluation of Connected and Automated Vehicles' Driving Loop: Node Weights and Driving Reliability

PONE-D-25-02537R1

Dear Dr. Zhang,

We’re pleased to inform you that your manuscript has been judged scientifically suitable for publication and will be formally accepted for publication once it meets all outstanding technical requirements.

Kind regards,

Zhengmao Li

Academic Editor

PLOS ONE

Additional Editor Comments (optional):

Reviewers' comments:

Reviewer's Responses to Questions

**Comments to the Author**

1. If the authors have adequately addressed your comments raised in a previous round of review and you feel that this manuscript is now acceptable for publication, you may indicate that here to bypass the “Comments to the Author” section, enter your conflict of interest statement in the “Confidential to Editor” section, and submit your "Accept" recommendation.

Reviewer #1: All comments have been addressed

Reviewer #2: All comments have been addressed

2. Is the manuscript technically sound, and do the data support the conclusions?

Reviewer #1: (No Response)

Reviewer #2: (No Response)

3. Has the statistical analysis been performed appropriately and rigorously? 

Reviewer #1: (No Response)

Reviewer #2: (No Response)

4. Have the authors made all data underlying the findings in their manuscript fully available?

Reviewer #1: (No Response)

Reviewer #2: (No Response)

5. Is the manuscript presented in an intelligible fashion and written in standard English?

Reviewer #1: (No Response)

Reviewer #2: (No Response)

6. Review Comments to the Author

Reviewer #1: (No Response)

Reviewer #2: (No Response)

7. PLOS authors have the option to publish the peer review history of their article (what does this mean? ). If published, this will include your full peer review and any attached files.

**Do you want your identity to be public for this peer review?** For information about this choice, including consent withdrawal, please see our Privacy Policy .

Reviewer #1: No

Reviewer #2: No

---

## [Editor Report · Acceptance letter]

PONE-D-25-02537R1

PLOS ONE

Dear Dr. Zhang,

I'm pleased to inform you that your manuscript has been deemed suitable for publication in PLOS ONE. Congratulations! Your manuscript is now being handed over to our production team.

Kind regards,

on behalf of

Dr Zhengmao Li

Academic Editor

PLOS ONE